

# Obesity, but not hypohydration, mediates changes in mental task load during passive heating in females

Aaron R. Caldwell[1], Jenna Burchfield[1], Nicole E. Moyen[1], Matthew A. Tucker[1,2], Cory L. Butts[1], R.J. Elbin[1] and Matthew S. Ganio[1]

[1] Exercise Science Research Center, University of Arkansas at Fayetteville, Fayetteville, AR, United States of America
[2] Georgia Prevention Institute, Augusta University, Augusta, GA, United States of America

## ABSTRACT

**Background**. The independent effects of hypohydration and hyperthermia on cognition and mood is unclear since the two stresses often confound each other. Further, it is unknown if obese individuals have the same impairments during hyperthermia and hypohydration that is often observed in non-obese individuals.

**Methods**. The current study was designed to assess the independent and combined effects of mild hypohydration and hyperthermia on cognition, mood, and mental task load in obese and non-obese females. Twenty-one healthy females participated in two passive heating trials, wherein they were either euhydrated or hypohydrated prior to and throughout passive heating. Cognition (ImPACT), mental task load (NASA-TLX), and mood (Brunel Mood Scale; BRUMS) were measured before and after a 1.0 °C increase in core temperature ($T_C$).

**Results**. After a 1.0 °C $T_C$ elevation, hypohydration resulted in greater ($p < 0.05$) body mass loss ($-1.14 \pm 0.48$ vs $-0.58 \pm 0.48$ kg; hypohydrated and euhydrated, respectively) and elevation in serum osmolality ($292 \pm 4$ vs $282 \pm 3$ mOsm; $p < 0.05$) versus euhydration. Hypohydration, independent of hyperthermia, did not affect mental task load or mood ($p > 0.05$). Hyperthermia, regardless of hydration status, impaired ($\sim$5 A.U) measures of memory-based cognition (verbal and visual memory), and increased mental task load, while worsening mood ($p < 0.05$). Interestingly, obese individuals had increased mental task load while hyperthermic compared to the non-obese individuals ($p < 0.05$) even while euhydrated. Hypohydration did not exacerbate any heat-related effects on cognition between obese and non-obese females ($p > 0.05$).

**Conclusion**. These data indicate that hyperthermia independently impairs memory-based aspects of cognitive performance, mental task load, and leads to a negative mood state. Mild hypohydration did not exacerbate the effects of hyperthermia. However, obese individuals had increased mental task load during hyperthermia.

Corresponding author
Matthew S. Ganio, msganio@uark.edu

## INTRODUCTION

Military personnel, construction workers, and firefighters are often exposed to occupational heat stress. These environments may cause hypohydration in addition to hyperthermia,

resulting in decrements to physical and cognitive performance (*Adam et al., 2008*). The decrements in cognitive performance may not only affect job productivity but may have serious implications for the safety of these workers. Therefore, understanding the independent and combined effects of hyperthermia and hypohydration on cognitive performance and mood may provide valuable information for those individuals who live and work in warm environments.

Hypohydration is commonly induced with heat stress and/or exercise because the elevation in core temperature induces fluid loss secondary to sweating. However, both heat stress and exercise can independently affect cognition and mood (*Tomporowski et al., 2007*; *Racinais, Gaoua & Grantham, 2008*). Therefore, a preferred methodology to independently examine the effect of hypohydration is fluid restriction. Some investigators have shown that mild hypohydration, with approximately 1–3% body mass loss by fluid restriction, causes slower psychomotor processing speed, impaired visual and spatial working memory, increased errors and skill impairments (*Wilson & Morley, 2003*; *Petri, Dropulic & Kardum, 2006*; *Patel et al., 2007*; *Suhr et al., 2010*; *Smith, Newell & Baker, 2012*; *Lindseth et al., 2013*). Additionally, mood ratings, such as feelings of tiredness and reduced vigor are observed with hypohydration (*Shirreffs, 2003*; *Szinnai et al., 2005*; *Patel et al., 2007*; *Pross et al., 2013*). However, another study reported no cognitive differences with mild hypohydration (∼2%) when fluid is restricted for 36-h (*Szinnai et al., 2005*).

Heat stress alone, even when hydration is maintained, can lead to cognitive impairments in more precise measures of cognitive function such as visual processing, pattern recognition memory, and spatial memory (*Wyon, Andersen & Lundqvist, 1979*; *Gaoua et al., 2011b*). Simpler tasks like measures of attention are preserved (*Gaoua et al., 2011b*). Many studies fail to report hydration status and/or maintain euhydration. Therefore, the independent effect of heat stress, and thus hyperthermia, is sometimes difficult to discern.

Increased body mass index (BMI) in healthy young individuals is correlated with impaired executive function, attention, problem solving, memory, and mental flexibility (*Gunstad et al., 2006*; *Gunstad et al., 2007*; *Fedor & Gunstad, 2013*). Although there is a paucity of data with obese individuals, some studies suggest obesity is linked with poorer cognitive performance compared to lean controls (*Boeka & Lokken, 2008*; *Lokken et al., 2009*; *Lokken et al., 2010*). Poor cognitive performance may be exacerbated in obese individuals when heat stressed because of impaired thermoregulatory control (*Bar-Or, Lundegren & Buskirk, 1969*; *Buskirk, Bar-Or & Kollias, 1969*; *Tucker et al., 2017*). However, the independent and combined effects of hyperthermia and hypohydration on cognition and mood in obese individuals have not been investigated. Obese individuals also appear to have alterations in brain structure (*Stanek et al., 2011*) and function (*Wolf et al., 2007*) which may affect changes in cognitive performance or mental task following certain stressors (*Mehta, 2015*). Further, given one-third of Americans are reported to be overweight or obese (*Ogden et al., 2015*), it is important to understand how cognition and mood are affected, not effected when obese individuals are heat stressed and/or hypohydrated.

Therefore, the purpose of this study was to examine the independent and combined effects of mild hypohydration (i.e., <2% body mass loss) and hyperthermia on cognition, mental task load, and mood in obese and non-obese individuals. We chose rather mild levels

**Table 1** Demographic data from both the obese and non-obese groups.

| M ± SD demographic data for non-obese and obese | | |
|---|---|---|
| | **Non-obese** | **Obese** |
| N | 11 | 10 |
| Age (y) | 22 ± 2 | 22 ± 2 |
| Height (cm) | 165 ± 5.8 | 161.3 ± 4.9 |
| Mass (kg) | 60.56 ± 5.76 | 79.79 ± 17.69[*] |
| Body Fat (%) | 25.13 ± 3.93 | 44.24 ± 5.08[*] |

**Notes.**

[*]Significantly different between groups ($p < 0.05$).

of hypohydration and hyperthermia because there is dearth of information in this area, but preliminary evidence suggests that mild hypohydration and hyperthermia may have cognitive (*Ganio et al., 2011*; *Armstrong et al., 2012*) and physiological (*Moyen et al., 2016*; *Tucker et al., 2018*) effects. Likewise, these mild levels are more likely to occur in everyday living. We hypothesized that mild hypohydration, induced by 24-h of fluid restriction, and hyperthermia would impair cognition and increase the mental effort necessary to complete a task, defined as mental task load (*Hart & Staveland, 1988*). Furthermore, we hypothesized that hypohydration would exacerbate hyperthermic-related impairments. We also hypothesized that impairments would coincide with a worsening in mood state (i.e., increased fatigue, lowered vigor, etc.). Finally, we hypothesized that obese individuals would have greater decrements both cognition, mental task load, and mood compared to non-obese participants.

## MATERIALS AND METHODS

Ten obese and 11 non-obese female adults aged 18 to 35 years old from the University of Arkansas and surrounding community were recruited for this study. A priori power calculations were not performed prior to data collection, but post-hoc power can be estimated from the statistics provided in the results. The University of Arkansas Institutional Review Board granted ethical approval to carry out this study within its facilities (IRB# 14-05-719). Prior to participation all participants signed a written informed consent. Participants were excluded if they had a history of any diseases that could affect body fluid balance, or if they had a history of psychiatric disease, concussion, dyslexia, or attention-deficit/hyperactivity disorder. This study focused on female participants considering the paucity of data within this population. Furthermore, participants were excluded if they took drugs influencing body fluid balance (e.g., diuretics) or regularly consumed the caffeine equivalent of greater than six cups of coffee each day. Those who had an irregular menstrual cycle, as well as those using an intrauterine device, or progesterone only form of birth control were excluded due to different hormone concentrations of estrogen and progesterone from normally menstruating females. Participant characteristics are in Table 1.

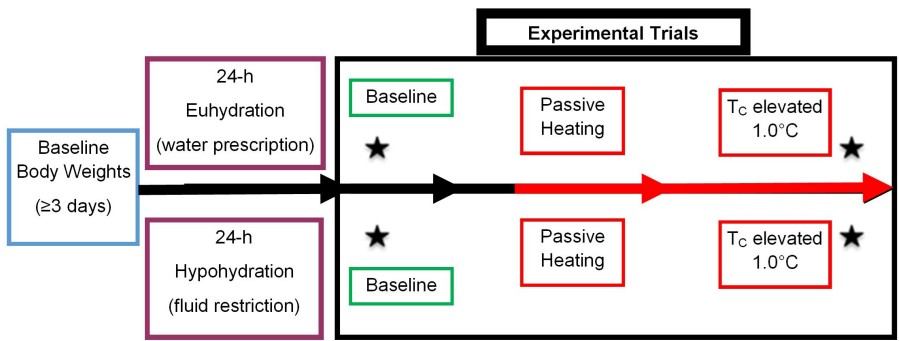

**Figure 1** **Schematic of experimental design.** Data collection began with 3 days of euhydrated body weights (blue boxes) collected every morning upon waking. In the 24-h prior to each trial participants either were assigned a fluid prescription protocol or withheld from fluid intake (purple boxes, middle left). Upon arriving to the lab on testing day, participants underwent baseline testing (* by green boxes within the black experimental trial box). Passive heating (red boxes) immediately began upon completion of baseline testing. Once a 1.0 °C increase in rectal temperature was reached post testing began (* by red boxes within black experimental trial box). Participants were removed from the environmental chamber after post testing was completed. *Represents when measures for perception mood (Brunel Mood Scale; BRUMS), mental task load (NASA-TLX), and cognitive performance (ImPACT) occurred.

## Experimental design

A randomized, repeated-measures design was used to determine the independent and combined effects of hypohydration and hyperthermia on cognitive performance, mental task load, and mood. Each participant completed a familiarization and two trials: one euhydrated and one hypohydrated. The familiarization visit occurred within two weeks prior to the first trial. Before each trial, participants participated in a 24-h hydration intervention (described below) before having their core temperature increased 1.0 °C via passive heating. Trials occurred during the follicular phase of the menstrual cycle (days 1–7). Menstrual cycle status was based on self-reported start of menses. A minimum of 48-h separated trials. The layout of each trial is in Fig. 1.

## Measures
### Brunel Mood Scale (BRUMS)
BRUMS is a validated instrument, derived from the Profile of Mood States, with 24-item Likert measures evaluating feelings of tension, depression, anger, vigor, fatigue, and confusion (*Terry et al., 1999*). The Likert type items ask respondents to indicate the extent to which they are experiencing these feelings. Items are anchored by "0 = Not at all", and "4 = Extremely". The reference timeframe utilized was "How you feel right now".

### Immediate Post-Concussion Assessment and Cognitive Testing (ImPACT)
The ImPACT computerized neurocognitive test evaluates performance by measuring six neurocognitive modules that assess memory, attention learning, processing speed and reaction time. Scores are created from the six neurocognitive modules that yield four composite scores; verbal memory, visual memory, visual motor processing speed, and reaction time in seconds. The ImPACT baseline test is a stable measure of neurocognitive

performance and minimizes practice effects through the use of multiple versions. ImPACT provides randomization of stimuli in alternate forms at the initial test session and all subsequent test sessions included different forms to minimize practice effects. Overall, current evidence would indicate that the ImPACT does adequately control for practice effects (*Schatz et al., 2006*).

### Mental task load

A visual analog scale (VAS) of concentration and the NASA Task Load (NASA-TLX) were utilized to quantify mental task load (*Hart & Staveland, 1988*; *Bijur, Silver & Gallagher, 2001*). The VAS is a continuous scale with two extreme statements used as anchors at opposite ends of a 100 mm line. The anchors used were "not at all strong(ly)" and "very strong(ly)". This scale was used to assess concentration using the statement "How hard did you have to concentrate to accomplish the tasks successfully?" The NASA-TLX utilized in this study measured subjective workload for mental demands, temporal demands, mental performance, effort, and frustration. The participant marks on a line, which represents a continuous scale for each of the previously mentioned sub-scales.

### Measures of hydration status

Four measures of hydration status were utilized in each experimental trial: body mass change, serum osmolality ($S_{OSM}$), urine osmolality ($U_{OSM}$), and urine specific gravity ($U_{SG}$).

Change in body mass was calculated from the baseline weight recorded as the average of the three or more weights obtained by the participant as follows. Participants were sent home with a scale (Model: IP-TITN-ZJXR, BalanceFrom) to record nude body weight at the same time, relative to the start time of their trials, each day for three consecutive days to establish a baseline body weight (*Cheuvront et al., 2004*). From this baseline, body weight, the amount of body mass lost, or gained, during the 24-h hydration intervention (see below) was calculated. The percentage of body weight change from baseline was used to determine percent hypohydration at the beginning of each trial (i.e., before heating but after the 24-h hydration intervention) and after each trial (i.e., after heating).

$U_{SG}$ was measured via a refractometer (clinical refractometer 30005, SPER Scientific, Scottsdale, AZ, USA). $U_{OSM}$ and $S_{OSM}$ were measured by freezing point depression (3D3 Advanced Instruments Osmometer, Model 3250, Norwood, MA, USA).

Previously determined cutoffs for body mass change (<1%), $S_{OSM}$ (<290 mOsm), $U_{OSM}$ (<700 mOsm), and $U_{SG}$ (<1.020) were used to define participants as euhydrated and participants were required to meet three of the four criteria prior to starting a trial (*Sawka et al., 2007*).

### Thermal measurements

Skin temperature ($T_{sk}$) was measured via skin thermocouples (Omega Engineering, Stamford, CT, USA) placed on the right anterior thigh (midway between the greater trochanter and lateral condyle), chest (midway between the axilla and areola), lateral calf (midway between the tibial condyle and malleolus), and upper arm. Mean-weighted $T_{sk}$ was calculated according to *Ramanathan (1964)*: $T_{sk} = 0.3(\text{chest} + \text{arm}) + 0.2(\text{thigh} + \text{calf})$.

A rectal thermister (Ret-1; Physitemp Inc., Clifton, NJ, USA) was used to measure core temperature ($T_C$). Mean body temperature ($T_b$) was calculated using $T_C$ and $T_{sk}$(19):
$T_b = 0.2(T_{sk}) + 0.8(T_C)$.

## Familiarization

Prior to experimental trials, participants underwent a familiarization visit. At this visit we recorded the participant's body mass via a scale (Health-o-Meter® digital scale, model 349KLX; Health-o-Meter, McCook, IL, USA), height via a stadiometer (model: 7701321004; Seca, Vogel & Hamburg, Germany), and body composition via dual x-ray absorptiometry (DXA, General Electric®, Lunar Prodigy Promo). Participants were classified as lean if their body fat was <32% whereas a body fat of >39% was classified as obese (*Gallagher et al., 2000*).

Participants were then familiarized with the BRUMS, ImPACT, the VAS for concentration, and the NASA-TLX. Participants were sent home with a scale to establish a baseline body weight as described above (*Cheuvront et al., 2004*).

## Experimental trials

For the 24-h prior to each experimental trial, participants underwent a hydration intervention. For the first trial participants kept diet and fluid logs and replicated the diet portion (i.e., calories) for the subsequent trial. During the euhydrated trial participants were encouraged to consume adequate fluid, including an extra 950 ml the night prior to the trial and 475 ml ∼2 h before the trial. Also, during passive heating for the euhydrated trial, participants consumed 37.5 °C water to maintain euhydration. For the hypohydrated trial participants were given a 250 ml bottle of water and asked to refrain from all other fluids and high moisture content foods (e.g., soup and fruits) for the 24-h prior to laboratory arrival. Furthermore, no fluid was ingested throughout passive heating.

Prior to each experimental trial participants refrained from alcohol and exercise for 24-h, food for 4-h, and caffeine for 8-h. After the 24-h fluid intervention, and upon arrival to the laboratory, pre-test compliance was verified with a 24-h history questionnaire. Participants then used a private bathroom to obtain a nude weight (Health-o-Meter® digital scale, model 349KLX; Health-o-Meter, McCook, IL, USA), insert a rectal thermistor, and provide a urine sample for measures of $U_{SG}$ and $U_{OSM}$. Participants were then fitted with a blood pressure cuff (Tango+; SunTech Medical, Inc., Morrisville, NC, USA) placed directly over the brachial artery to obtain blood pressure via electrosphygmomanometry, and a monitor around the chest continuously recording heart rate (HR; Polar Electro Inc., Lake Success, NY, USA). Skin thermocouples were then placed on the appropriate locations (see above).

Participants were then dressed in a water-perfused, tube-lined suit that covers the entire body, except the head, face, hands, and feet (Allen-Vanguard Technologies, Ottawa, Canada). The suit permits the control of $T_{SK}$ and $T_C$ temperature by changing the temperature of the water perfusing the suit. After instrumentation and prior to heating, participants completed BRUMS mood assessment, ImPACT and mental task load measures (VAS and NASA-TLX). Then an intravenous catheter was inserted into the antecubital vein. After participants laid supine for ∼30 min a blood sample was obtained. During the resting

period water at 34 °C was perfused through the suit. After this resting period, participants were then exposed to passive heat stress by perfusing 45 °C water through the suit and moved into an environmental chamber ($34.5 \pm 0.6$ °C, $30.8 \pm 0.9\%$ relative humidity). HR, $T_{SK}$, $T_B$ and $T_C$ were continuously measured via computer software (LabChart7, Colorado Springs, CO). Once a 1.0 °C $T_C$ increase was achieved, $T_C$ was maintained at this temperature by stabilizing the temperature of the water going through the suit. At the 1.0 °C increase in $T_C$, a blood sample was obtained. Participants then completed the BRUMS mood assessment, ImPACT cognitive test and mental task load measures (VAS and NASA-TLX). All ImPACT test sessions included different forms to minimize practice effects. Once testing was completed, nude body mass, to calculate total body water loss, was obtained and participants provided another urine sample.

## Statistical analysis

Statistical analyses were performed using IBM SPSS Statistics v23. In order to establish evidence for the experimental design, a series of separate univariate ANOVAs were performed for all physiological ($T_C$, $T_B$, $T_{SK}$, HR), hydration ($U_{SG}$, $U_{OSM}$, $S_{OSM}$, and body mass change) variables.

In order to test our hypotheses, three two (hypohydrated vs euhydrated) by two (baseline vs hyperthermia) by two (obese vs non-obese) multivariate repeated measures analysis of variance (MANOVAs) were performed on cognitive performance, mental task load, and mood outcome variables. *Cognitive performance* outcome scores were derived from the ImPACT battery and included verbal memory, visual memory, processing speed, and reaction time. *Mental task load* scores were derived from the NASA-TLX and VAS and included mental demands, temporal demands, mental performance, effort and frustration, and concentration. *Mood* scores were derived from the BRUMS and included tension, depression, anger, vigor, fatigue, and confusion.

An alpha of less than 0.05 was deemed significant. In the case of a multivariate alpha of less than 0.10 an exploratory analysis of the univariate effects were performed. If a statistically significant multivariate effect was identified, then corresponding univariate ANOVAs were examined.

# RESULTS

## Confirmation of experimental design

### Hydration

Overall, there were no differences between hydration variables in the obese and non-obese groups. In the euhydrated trial, subjects arrived with a body mass loss of only $0.02 \pm 1.19\%$ (relative to 3-day euhydrated baseline body mass), a $S_{OSM}$ of $283 \pm 2$ mOsm, $U_{OSM}$ of $331 \pm 198$ mOsm, and a $U_{SG}$ of $1.011 \pm 0.005$. $S_{OSM}$ did not increase in the euhydrated trial with hyperthermia (mean change: $-1.8$ mOsm $\pm 3.5$; $p = 0.01$). There was a small change in body mass (mean change: $-0.64 \pm 0.56\%$; $p = 0.05$). $U_{OSM}$ and $U_{SG}$ increased slightly after hyperthermia during the euhydrated trial (mean change: $52 \pm 199$ mOsm and $0.003 \pm 0.006$, respectively; $p > 0.05$). However, participants were still considered euhydrated by current guidelines (*Armstrong, 2007*).
**Table 2  Hydration data collected pre- and post-heating while euhydrated and hypohydrated in obese and non-obese individuals.**

| Hydration markers | | Non-obese | Obese |
|---|---|---|---|
| Body mass loss (kg) from 3-day euhydrated baseline | | | |
| Pre-heating | Euhydrated | $0.62 \pm 1.08$ | $-0.44 \pm 1.39$ |
| | Hypohydrated | $-0.97 \pm 0.74^b$ | $-0.91 \pm 0.90^b$ |
| Additional body mass loss (kg) from passive heating | | | |
| Post-heating | Euhydrated | $-0.89 \pm 0.43$ | $-0.23 \pm 0.24$ |
| | Hypohydrated | $-1.46 \pm 0.29^b$ | $-0.78 \pm 0.37^b$ |
| Urine specific gravity | | | |
| Pre-heating | Euhydrated | $1.008 \pm 0.005$ | $1.009 \pm 0.006$ |
| | Hypohydrated | $1.024 \pm 0.004^b$ | $1.026 \pm 0.004^b$ |
| Post-heating | Euhydrated | $1.009 \pm 0.003$ | $1.011 \pm 0.007$ |
| | Hypohydrated | $1.026 \pm 1.004^{a,b}$ | $1.029 \pm 0.004^{a,b}$ |
| Urine osmolality (mOsm) | | | |
| Pre-heating | Euhydrated | $347 \pm 195$ | $377 \pm 226$ |
| | Hypohydrated | $961 \pm 105^b$ | $1051 \pm 116^b$ |
| Post-heating | Euhydrated | $401 \pm 104$ | $454 \pm 271$ |
| | Hypohydrated | $992 \pm 89^{a,b}$ | $1126 \pm 114^{a,b}$ |
| Serum osmolality (mOsm) | | | |
| Pre-heating | Euhydrated | $285 \pm 2$ | $283 \pm 3$ |
| | Hypohydrated | $289 \pm 4^b$ | $287 \pm 3^b$ |
| Post-heating | Euhydrated | $284 \pm 3$ | $281 \pm 2$ |
| | Hypohydrated | $294 \pm 4^{a,b}$ | $291 \pm 3^{a,b}$ |

**Notes.**

[a] Significant difference from corresponding pre-heating value.
[b] Significant difference from corresponding euhydration value.

In the hypohydrated trial, percent body mass loss ($-0.94 \pm 0.76\%$), $S_{OSM}$ ($287 \pm 3$ mOsm), $U_{OSM}$ ($986 \pm 118$ mOsm), and $U_{SG}$ ($1.025 \pm 0.004$) were all elevated relative to the euhydrated trial (all $p < 0.01$). During the hypohydrated trial, body mass, $U_{OSM}$, $U_{SG}$, and $S_{OSM}$ had additional changes with passive heating ($-1.15 \pm 0.45\%$, $53 \pm 50$ mOsm, $0.002 \pm 0.003$, and $-4.8 \pm 2.7$ mOsm, respectively; all $p < 0.05$). When hyperthermic and hypohydrated, $U_{OSM}$, $U_{SG}$, $S_{OSM}$, and body mass loss were greater compared to euhydrated and hyperthermic (all $p < 0.001$). Detailed hydration data are available in Table 2.

### Physiological responses

As expected, $T_C$, $T_{SK}$, and $T_B$ increased during hyperthermia (mean change: $0.97 \pm 0.05\,°C$, $2.65 \pm 1.00°\,C$, and $1.14 \pm 0.09\,°C$, respectively; all $p < 0.001$). However, none of the other physiological variables (i.e., $T_C$, $T_{SK}$, and $T_B$) were significantly different between non-obese and obese groups in the euhydrated trial (all $p > 0.05$).

Hypohydration did not alter baseline $T_C$, $T_B$, or HR with the exception of $T_{SK}$ being higher (mean difference: $0.37 \pm 0.54\,°C$; $p < 0.01$; independent effect of hydration). HR was significantly higher in obese compared to non-obese at baseline in the hypohydrated trial (mean difference: $10 \pm 18$ bpm; $p = 0.02$). Also, obese females had slightly higher HR

**Table 3  Physiological data collected pre- and post-heating while euhydrated and hypohydrated in obese and non-obese individuals.**

| Physiological markers | | Non-obese | Obese |
|---|---|---|---|
| Rectal temperature (°C) | | | |
| Pre-heating | Euhydrated | $36.77 \pm 0.28$ | $36.78 \pm 0.41$ |
| | Hypohydrated | $36.83 \pm 0.24$ | $36.87 \pm 0.43$ |
| Post-heating | Euhydrated | $37.72 \pm 0.26$[a] | $37.76 \pm 0.39$[a] |
| | Hypohydrated | $37.82 \pm 0.24$[a] | $37.85 \pm 0.44$[a] |
| Skin temperature (°C) | | | |
| Pre-heating | Euhydrated | $35.10 \pm 0.51$ | $34.97 \pm 0.45$ |
| | Hypohydrated | $35.37 \pm 0.33$ | $35.43 \pm 0.58$ |
| Post-heating | Euhydrated | $37.62 \pm 0.68$[a] | $37.75 \pm 0.77$[a] |
| | Hypohydrated | $37.90 \pm 0.62$[a,b] | $38.27 \pm 0.80$[a,b] |
| Mean body temperature (°C) | | | |
| Pre-heating | Euhydrated | $36.60 \pm 0.26$ | $36.60 \pm 0.39$ |
| | Hypohydrated | $36.80 \pm 0.23$ | $36.72 \pm 0.42$ |
| Post-heating | Euhydrated | $37.71 \pm 0.27$[a] | $37.76 \pm 0.41$[a] |
| | Hypohydrated | $37.83 \pm 0.24$[a,b] | $37.90 \pm 0.44$[a,b] |
| Heart rate (bpm) | | | |
| Pre-heating | Euhydrated | $64 \pm 5$ | $70 \pm 11$[c] |
| | Hypohydrated | $65 \pm 8$ | $75 \pm 10$[c] |
| Post-heating | Euhydrated | $94 \pm 8$[a] | $104 \pm 14$[a,c] |
| | Hypohydrated | $98 \pm 8$[a] | $106 \pm 11$[a,c] |

**Notes.**

[a] Significant difference from corresponding pre-heating value.
[b] Significant difference from corresponding euhydration value.
[c] Significant difference from corresponding non-obese value.

after passive heating in the hypohydrated trial (mean difference: $9 \pm 19$ bpm; $p = 0.04$). There were no other differences in the physiological variables (i.e., $T_C$, $T_{SK}$, and $T_B$) between non-obese and obese groups in the hypohydrated trial (all $p > 0.10$).

In the hypohydrated trial, passive heating increased $T_C$ $0.99 \pm 0.02$ °C, $T_{SK}$ $2.69 \pm 1.01$ °C, $T_B$ $1.16 \pm 0.11$ °C, and HR $32 \pm 8$ bpm (all $p < 0.01$). $T_{SK}$ and $T_B$ were greater at the end of the hypohydrated trial compared to the euhydrated trial (mean difference: $0.40 \pm 0.69$ and $0.13 \pm 0.23$ °C, respectively; $p = 0.01$). $T_C$ did not differ between trials at the end of heating ($p > 0.05$). Detailed physiological data are available in Table 3.

## Testing of hypotheses
### Cognitive performance
At a multivariate level, there was not a three-way interaction between hyperthermia, hydration, group and obesity on cognitive performance $F(4, 16) = 1.31$, $p = 0.30$, Wilks' $\Lambda = 0.25$. Also, there was not a two-way multivariate interaction between obesity and hyperthermia, $F(4, 16) = 0.22$, $p = 0.92$, Wilks' $\Lambda = 0.95$, or obesity and hydration $F(4, 16) = 0.20$, $p = 0.92$, Wilks' $\Lambda = 0.95$.

Further, there was no multivariate interactions between hydration and hyperthermia, $F(4, 16) = 1.53$, $p = 0.24$, Wilks' $\Lambda = 0.75$. However, there was an independent, multivariate effect of hyperthermia indicating a decrease in cognitive performance, $F(4, 16) = 3.01$, $p = 0.049$, Wilks' $\Lambda = 0.57$. Follow-up univariate analyses, presented in Fig. 2, indicated that hyperthermia led to a reduction in verbal recognition memory, $F(1, 19) = 6.98$, $p = 0.02$, and trend towards reduction in visual working memory $F(1, 19) = 3.58$, $p = 0.07$, but no change in visual processing speed or reaction time. Further, there was trend towards an independent multivariate effect of hydration status $F(4, 16) = 2.68$, $p = 0.07$, Wilks' $\Lambda = 0.59$. Follow-up univariate tests revealed that only visual working memory was slightly reduced when hypohydrated (mean change: $-4.9 \pm 1.8$ A.U., $p = 0.01$).

### Mental task load

There were no multivariate interactions between obesity, hydration and hyperthermia on mental task load $F(6, 14) = 0.74$, $p = 0.63$, Wilks' $\Lambda = 0.24$. Further, there was not a multivariate interaction between obesity and hydration status $F(6, 14) = 1.164$, $p = 0.38$, Wilks' $\Lambda = 0.67$. However, there was an interaction between obesity and hyperthermia $F(6, 14) = 2.93$, $p = 0.04$, Wilks' $\Lambda = 0.44$. Interestingly, only in obese participants, hyperthermia increased mental demand ($p < 0.04$), decreased ratings of performance ($p < 0.01$), and increased effort ($p < 0.01$; Fig. 3). Pairwise comparisons indicate that temporal demand and frustration were unaffected by hyperthermia in either group (Figs. 3B and Figs. 3B; $p > 0.05$). In both obese and non-obese participants concentration increased (Fig. 3F; $p < 0.05$).

There was trend for a multivariate interaction between hydration status and hyperthermia, $F(6, 14) = 2.93$, $p = 0.06$, Wilks' $\Lambda = 0.46$. Exploratory analysis of the univariate tests only revealed differences in temporal demand, which pairwise comparisons indicated a small but significant increase in temporal demand when hypohydrated (mean change: $5.5 \pm 9.1$ A.U.; $p < 0.05$), but not when euhydrated (mean change: $1.3 \pm 7.5$ A.U.; $p = 0.42$). Independent of hyperthermia, there was no multivariate effect of hypohydration, $F(6, 14) = 2.93$, $p = 0.29$, Wilks' $\Lambda = 0.63$.

### Mood state

There was no interaction between obesity, hydration, and hyperthermia, $F(6, 14) = 1.51$, $p = 0.25$, Wilks' $\Lambda = 0.61$. Also, there was no interaction between hydration status, $F(6, 14) = 1.73$, $p = 0.19$, Wilks' $\Lambda = 0.57$, or hyperthermia, $F(6, 14) = 0.74$, $p = 0.63$, Wilks' $\Lambda = 0.76$ (i.e., non-significant interaction). Furthermore, there was no interaction between hydration and hyperthermia on mood $F(6, 14) = 0.95$, $p = 0.49$, Wilks' $\Lambda = 0.711$. Mood state did not appear to change when hydration status was altered, independent of hyperthermia, $F(6, 14) = 1.98$, $p = 0.14$, Wilks' $\Lambda = 0.54$.

However, there was trend for a main effect of hyperthermia $F(6, 14) = 2.79$, $p = 0.05$, Wilks' $\Lambda = 0.46$. An exploratory univariate analysis, displayed in Fig. 4, indicated increases in anger (mean change: $0.4 \pm 0.46$ A.U.; $p < 0.01$) and depression (mean change: $-0.9$

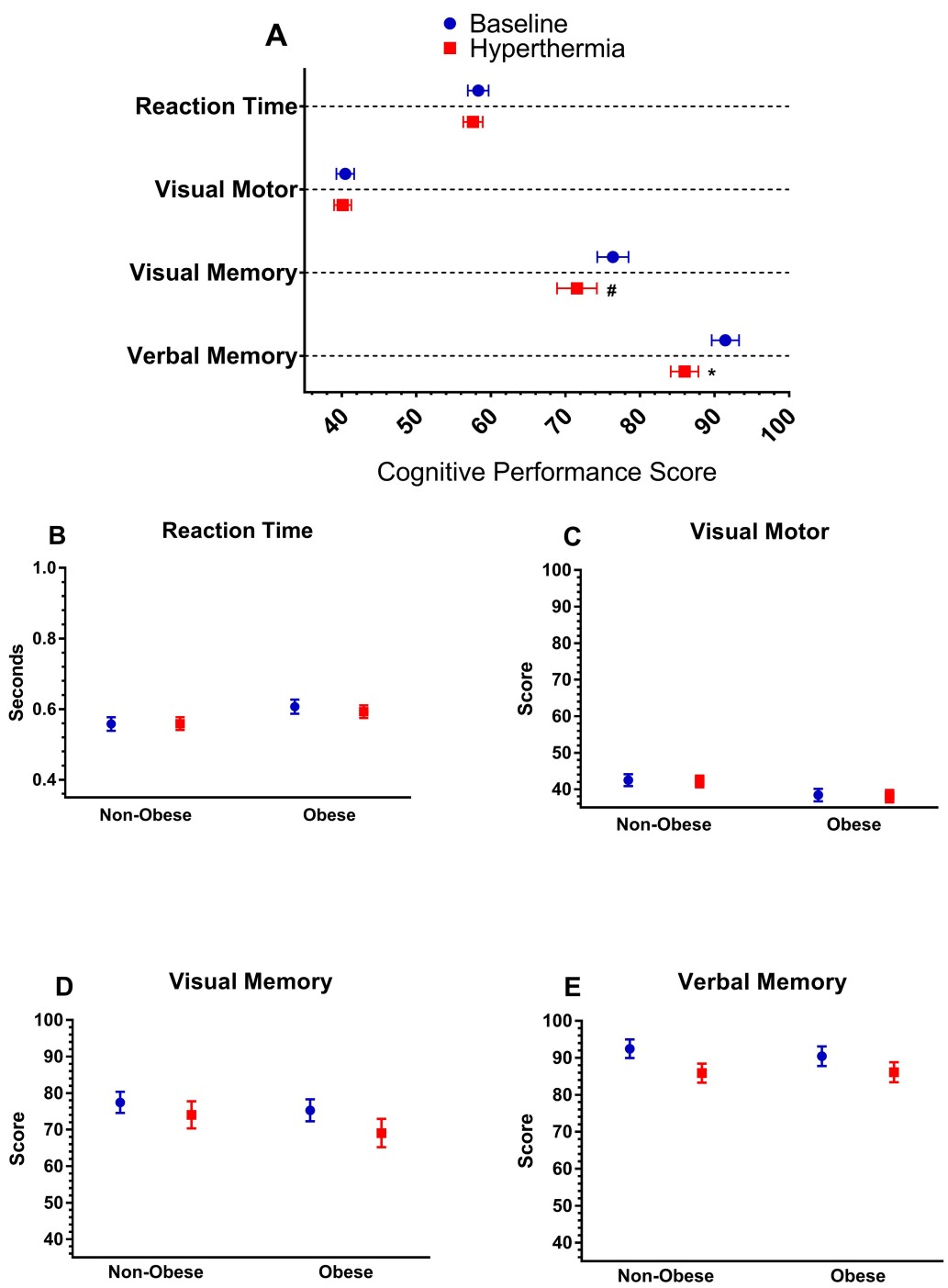

**Figure 2 Cognitive performance.** (A) Cognitive performance variables ($y$-axis) from the ImPACT at baseline and during hyperthermia. Below are (B) reaction time (C) visual motor performance, (D) visual memory, and (E) verbal memory from the ImPACT for each group (obese and non-obese) at baseline and during hyperthermia. *Indicates significant difference from baseline ($p < 0.05$), # Indicates a trend for a difference between baseline ($p < 0.10$). Symbols in (A) are vertically offset for clarity.

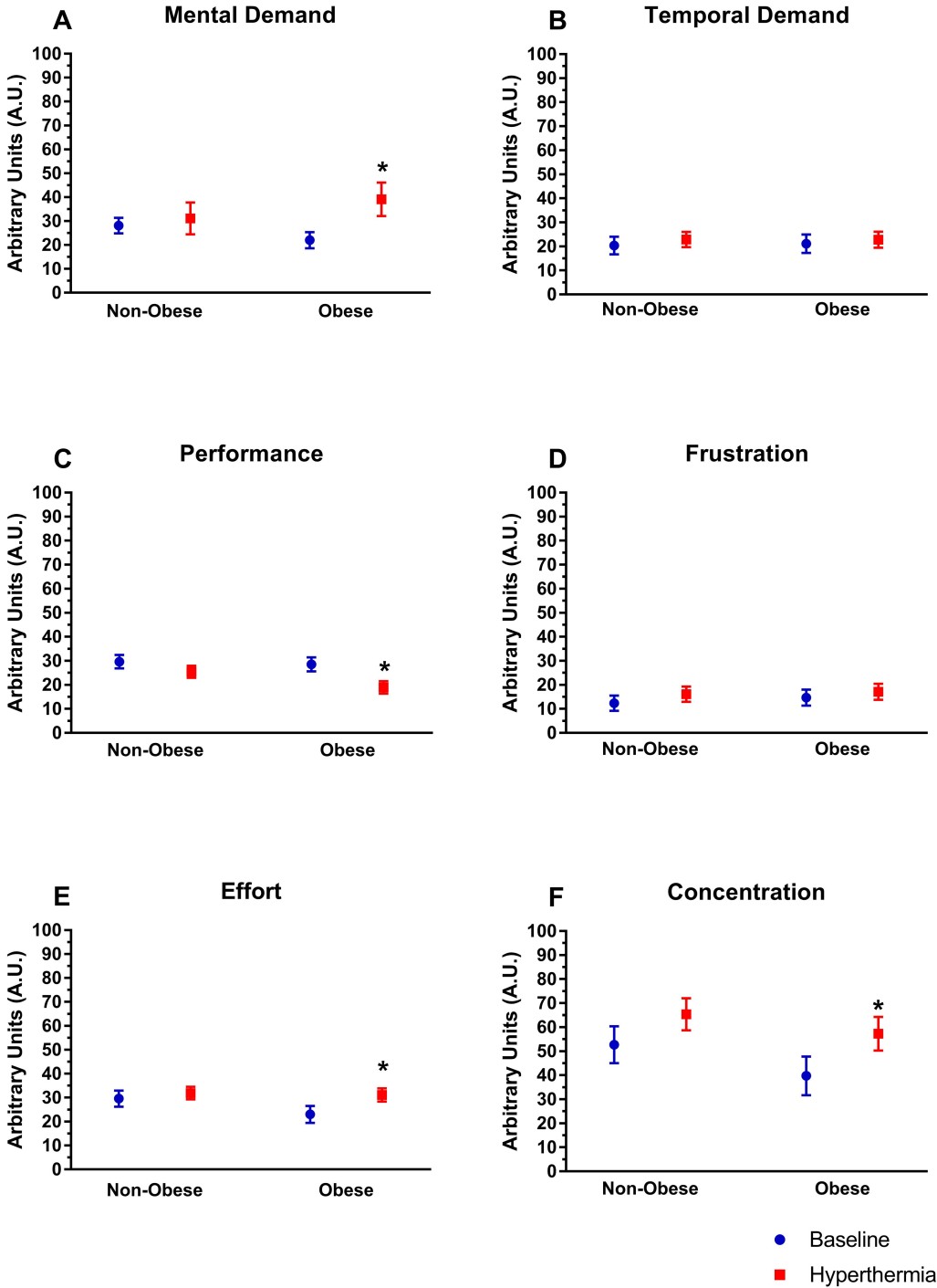

**Figure 3 Mental task load.** Perceived mental task load (NASA-TLX, visual analog score in arbitrary units) for (A) mental demand, (B) temporal demand (C) performance, (D) frustration, (E) effort, and (F) concentration for each group (obese and non-obese) at baseline and during hyperthermia. An asterisk indicates significant difference from baseline within group ($p < 0.05$).

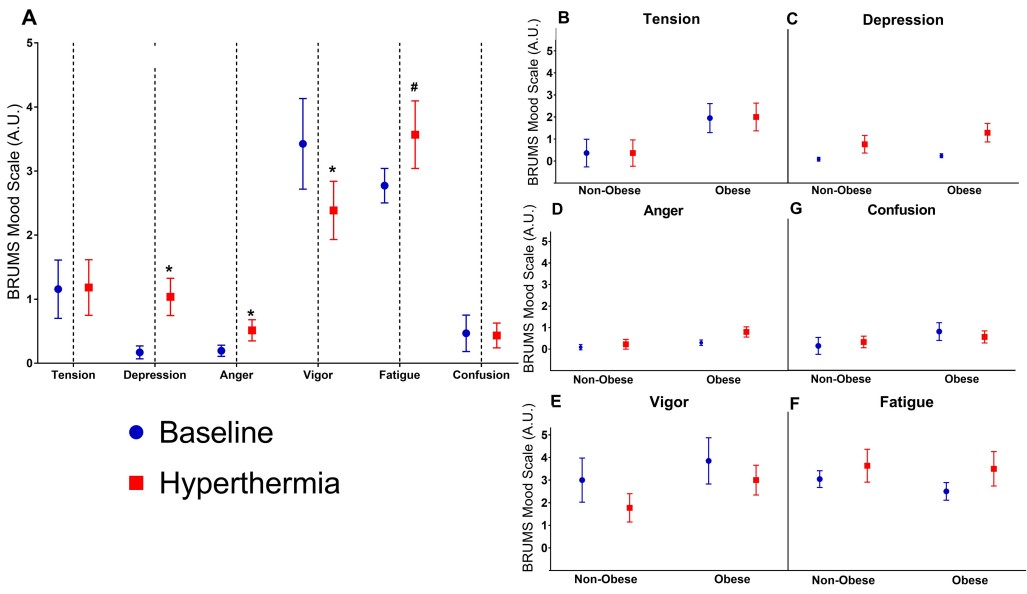

**Figure 4  Mood state scores from the Brunel Mood Scale.** (A) Perceived mood state scores (in arbitrary units) from the Brunel Mood Scale (BRUMS) compared at baseline and during hyperthermia. To the right are the mood responses split by group (obese versus non-obese) including (B) tension, (C) depression, (D) anger, (E) vigor, (F) fatigue, and (G) confusion. An asterisk indicates significant difference from baseline ($p < 0.05$), # indicates a trend for a difference between baseline ($p < 0.10$).

A.U. $\pm$ 1.1; $p < 0.01$), and decreased vigor (mean change: $-1.1 \pm 2.1$ A.U.; $p = 0.03$) during hyperthermia. There were no changes in tension, fatigue, or confusion (all $p > 0.10$; Fig. 4).

## DISCUSSION

In this study, we examined the independent and combined effects of mild hypohydration and hyperthermia on cognition, mental task load, and mood between obese and non-obese females. Mild hypohydration did not influence cognitive performance, mental task load, or mood state regardless of hyperthermia or obesity. Hyperthermia, when observed separately from hydration status (i.e., main effect of hyperthermia), decreased cognitive performance. In particular, measures of verbal memory and visual memory were impaired, but reaction time and visual motor processing were maintained. At the same time, most measures of mental task load (demand, performance, effort, and concentration) were increased during hyperthermia. This analysis included obese and non-obese individuals. Follow-up analysis revealed that much of the effect was driven by an independent effect of the obese individuals having increased mental task load during hyperthermia; for the most part, the non-obese individuals had no measurable change. Lastly, mood state, while unaffected by obesity or hydration, was negatively affected by hyperthermia, which corresponded to increased feelings of depression, anger, and reduced vigor. Overall, hyperthermia, but not hypohydration, decreases cognitive performance, and worsens mood state in young

females. Mental task load specifically increased in obese females while hyperthermic; non-obese female's mental task load was unaffected by hyperthermia.

The rationale for the current study stems from Baar's global workspace theory of the conscious mind which has a limited capacity (*Baars, 1993*). Baar's suggests cognitive processes are working in competition with each other rather than in parallel. Selected activities dominate cognitive awareness by harnessing executive function. Stressors could increase the need for cognitive function in one area of the brain, compromising other areas of the brain via cognitive load competition (*Cohen & Mermelstein, 1983*; *Baars, 1993*). In our study, hyperthermia impaired cognitive performance while increasing perceived symptoms (thermal sensation) and impairing mood state. Therefore, competition surfaced in executive function, and in order to preserve the cognitive skill being used, the brain exerts more effort toward the skills, sparing cognitive performance impairment in reaction time and visual processing. Thus, in order to meet the demands for stable cognitive performance, mental task load increases.

## Effects of mild hypohydration

In the current study, cognitive function, mood state, and mental task load were preserved with ~1% hypohydration in both obese and non-obese females. Like in previous studies (*Amos et al., 2000*; *Grego et al., 2005*), the magnitude of hypohydration may not have been sufficient to induce an observable impairment in cognitive function. Theoretically, there is a threshold at which the mental task load becomes too great, leads to exhaustion, and cognitive function beings to decline. Young healthy participants are able to adapt to progressive body water deficit, even up to 2.6% body mass loss, and maintain cognitive performance (*Szinnai et al., 2005*), which is in agreement with the current study. *Kempton et al. (2011)* suggest this phenomenon is fueled by the increase in the fronto-parietal brain activation resulting in unimpaired cognitive scores and a deteriorated mental task load. During fluid restriction there may be an initial drop in short-term memory followed by a return to baseline 3.5 h later, independent of rehydration (*Cian et al., 2000*; *Szinnai et al., 2005*). In our study, no changes in cognitive function during hypohydration, via ImPACT, were detected. Overall, cognitive function appear to be well-preserved with mild hypohydration during fluid restriction.

## Effects of hyperthermia

In two separate trials, we had subjects arrive either euhydrated or hypohydrated, and subjects maintained this status throughout our heating protocol (Table S1). Many cognitive performance variables were stable when hyperthermic. However, verbal memory, and to a lesser degree visual memory, were impaired (Figs. 2 and 3). These visual and verbal memory impairments are similar to those observed in previous studies (*Gaoua et al., 2011b*; *Schlader et al., 2013*). Cognitive function involving greater neuronal resources are impaired to a greater extent during heat stress than less demanding tasks (*Hocking et al., 2001*; *Simmons et al., 2008*; *Gaoua et al., 2011a*; *Gaoua et al., 2011b*; *Schlader et al., 2015*) and memory is a more complex task than attention tasks. Baddedly's working memory theory states verbal memory is controlled by the phonological loop, while the "visuospatial sketchpad" controls

visual memory (*Baddeley & Hitch, 1974*). These functions are connected and controlled by central executive function. In order to store memory as long-term memory, rehearsal is necessary, and visual memory is of limited capacity. Therefore, if the brain is exerting effort toward storing verbal memory, which is an easier task, competition occurs and the brain's ability to store visual memory, a more complicated task, is limited.

Unlike memory, reaction time and attention were maintained during heat stress (Fig. 2). Other research suggests heating increases arousal, and thus maintains reaction time and attention, until a critical $T_C$ is reached where impairments occur, which our participants were well below (*Hancock & Vasmatzidis, 2003*). Participants in our study also reported difficulty concentrating (~33% increase in concentration) during hyperthermia. However, attention, as measured by reaction time and motor processing speed was not impaired. Difficulty concentrating could impair the ability to form long-term memories and affect the learning ability of young females during heat stress. Therefore, cognitive impairments seen with hyperthermia in this study could be influenced by both heat-associated physiological responses, and distractions formed from increased symptoms and a negative mood state.

## Influence of hypohydration on responses to hyperthermia

Although some cognitive performance variables were impaired during hyperthermia, they were not impaired to a greater degree when mild hypohydration occurred. Given that our sample was young and healthy, physiological mechanisms could compensate in order to stabilize variables after a certain amount of stress. Previous research has suggested that a longer period of hypohydration (60-h of fluid restriction) in young individuals did not exacerbate impairments seen at lower levels of hypohydration, but instead young individuals were able to maintain cognitive performance (*Szinnai et al., 2005*). Other models of inducing hypohydration, such as diuretics or exercise, may be necessary to investigate if a greater degree of hypohydration affects responses to hyperthermia.

## Influence of obesity

Interestingly, there was no effect of body composition on cognitive performance, mood, or presentation of any symptoms. However, mental task load, in many areas, was significantly affected in the obese, but not the non-obese participants during hyperthermia. This indicates that obese individuals had a perception that the heat was imposing a greater demand upon them. There is some evidence that autonomic dysfunction associated with obesity increases fatigue development in obese adults during periods of high mental stress (*Mehta, 2015*). In particular, it is concerning that decrements in physical performance occur much quicker in obese adults during periods of high mental stress (*Mehta, 2015*). The stress of hyperthermia was not sufficient to impair cognitive performance, symptoms, or mood with moderate heating any more than it did for non-obese participants. The obese individuals did experience increased cardiovascular strain (as evidence by increased HR in this study), but the ability to maintain cerebral perfusion during cognitive task between groups is probably unaffected (*Bar-Or, Lundegren & Buskirk, 1969*). In summary, it is worth noting that performance is maintained in obese individuals, but they find it harder to perform cognitive tasks while in the heat.

## Limitations

Mood score only reflected negative feelings and did not give an overall mood state, which may reduce our ability to fully interpret the overall mood state of the females. Also, mood assessment occurred after instrumentation with the rectal probe which may negatively affect mood state. Furthermore, the cognitive assessment (ImPACT) used was a series of five modules. The modules are always administered in the same order for each test. Thus, the cognitive domains measured were not counterbalanced. The lack of counterbalancing could indicate impairment to one domain might affect a later module and the corresponding domain. However, given the conditions of the experiment were counterbalanced, we are confident in our ability to examine the effect of hydration and hyperthermia status on the measured variables. Additionally, we recognize impairment to attention might impair latter modules. Although, most studies using heat stress indicate attention is preserved, studies using hypohydration suggest attention might deteriorate.

Furthermore, since we were using a controlled passive heating protocol involving a water-perfusion suit, heat was administered to the skin first and subsequently caused an increase in $T_C$. The extension that this form of hyperthermia results in changes in cognition relies on the assumption that changes in $T_C$ (in this case rectal) reflect changes in brain temperature (*Covaciu et al., 2011*). The water-perfusion suits in this study do create an unnatural hyperthermic environment surrounding the skin with 49 °C water. This allows for rapid increases in $T_{SK}$ and $T_C$, but limits external validity because such an extreme environment is unlikely to exist in an occupational setting. Also, a number of other factors occur during exercise or physical labor, such as blood flow demands and heat production at the muscle that may influence the cognitive response to exertional heat stress. Therefore, these results may not be generalized to exertional hyperthermia.

## CONCLUSIONS

In this study, we demonstrated that apparently healthy adult females experience mild cognitive impairments, specific to memory, during moderate hyperthermia (+1.0 °C increase in $T_C$). These impairments are not worse in obese individuals nor were cognitive impairments worsened when hypohydrated. Hypohydration did not affect any variable related to cognitive function, with the exception of a trend in visual memory. Impairments in mood, evidenced by increased fatigue and depression, and mental task load accompanied cognitive impairments during hyperthermia. Obese individuals experienced a greater demand of the mental activities during hyperthermia, but did not experience greater cognitive impairments. Overall, obese individuals may not experience greater cognitive impairments during hyperthermia, but they likely have to put forth a greater effort to maintain their cognitive abilities. Therefore, during periods of high heat stress, obese individuals are unlikely to experience cognitive performance decrements despite exerting greater effort in comparison to their lean peers. Future studies should seek to determine if physical tasks are equally affected by hyperthermia and hypohydration in obese and non-obese adults.

# ACKNOWLEDGEMENTS

We would like to thank everyone at the Exercise Science Research Center for helping facilitate this research. We would like to thank Shari Witherspoon and J.D. Adams for their help with the administrative work in the laboratory. Further, we would like to thank Melissa Anderson, Megan Rosa-Caldwell, and Dr. Sean Mulvenon for their help reviewing this manuscript and statistical analyses.

### Funding

This work was supported by the corresponding author's internal money from the University of Arkansas and a grant from the University of Arkansas' Honors College. The funders had no role in study design, data collection and analysis, decision to publish, or preparation of the manuscript.

### Grant Disclosures

The following grant information was disclosed by the authors:
University of Arkansas.
University of Arkansas' Honors College.

### Competing Interests

The authors declare there are no competing interests.

### Author Contributions

- Aaron R. Caldwell analyzed the data, prepared figures and/or tables, authored or reviewed drafts of the paper, approved the final draft.
- Jenna Burchfield, Nicole E. Moyen and Matthew A. Tucker conceived and designed the experiments, performed the experiments, contributed reagents/materials/analysis tools, approved the final draft.
- Cory L. Butts performed the experiments, contributed reagents/materials/analysis tools, authored or reviewed drafts of the paper, approved the final draft.
- R J. Elbin conceived and designed the experiments, analyzed the data, contributed reagents/materials/analysis tools, authored or reviewed drafts of the paper, approved the final draft.
- Matthew S. Ganio conceived and designed the experiments, analyzed the data, authored or reviewed drafts of the paper, approved the final draft.

### Human Ethics

The following information was supplied relating to ethical approvals (i.e., approving body and any reference numbers):

The University of Arkansas Institutional Review Board granted Ethical approval to carry out this study (approval number: 14-05-719).
## Data Availability

The raw data are provided in the Supplemental Files.

## Supplemental Information

Supplemental information for this article can be found online at http://dx.doi.org/10.7717/peerj.5394#supplemental-information.

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
