# Peer review of "Obesity, but not hypohydration, mediates changes in mental task load during passive heating in females"

_PeerJ, doi:10.7717/peerj.5394_

## Round 0.1 · original submission · Major Revisions

Dear authors

Sorry for the delayed review process. It was a bit difficult to find experts with the right time to revise your paper. Now we have received comments from three reviewers. I would like to invite you to carefully revise each point indicated by the reviewers and send your revised manuscript. Please make sure you send a detailed response letter to the points of discussion indicated and also highlight every change and edit in the revised version of your manuscript.

Reviewer 1 ·

Basic reporting

Tables 2, 3 and 4 don`t present p values and they should, since they intend to demonstrate the differences between analyzed groups and situations. The figures need explanatory footnote, however, I would consider deleting them, because they repeat data already described both in the tables and the text.

Experimental design

No comment.

Validity of the findings

No comment.

Additional comments

There are some phrases along the manuscript that lack the initial capital letter (lines 52 and 351). Some terms appear unabbreviated again, after their acronyms have already been explained (lines 259 and 260). In the discussion (line 353) Kempton et al suggest (not "suggests" since et al means and others). In the conclusion (line 433) the word "to" is missing...effort TO maintain.

Reviewer 2 ·

Basic reporting

No comment.

Experimental design

No comments.
The experimental design is clear and seems appropriate to respond the objective of the present study. The main question is why authors choose hypohydration equal to 2 percent normal body mass and hyphertemia equal 1o C to compare differences in cognition, mental task load and mood between obese and non-obese female. What physiological or perceptual response could impair or improve these variables differently, based on a heterogeneous group in body composition. This may not be a weakness of this study, but it might be included in the rationale.

Validity of the findings

No comments.

Additional comments

Manuscript entitled “Obesity, but not hypohydration, mediates changes in mental task load during passive heating in females” (#25543)

General comments 

This manuscript makes an important contribution to the literature in the area by providing information regarding the effects of hypohydration and hyperthermia on cognition, and mood in obese and non-obese. The objective of this manuscript, as described by authors, was to examine the independent and combined effects of mild hypohydration (i.e., <2% body mass loss) and hyperthermia on cognition, mental task load, and mood in obese and non-obese individuals. My suggestion is that the introduction and discussion must be reviewed to include physiological mechanisms hypothesis for the results found in the present study, mainly related to obesity.

Comments to the Author

1) Hydration: The authors chose to examine the impact of an approximate dehydration level equal to 2 percent normal body mass.  A number of manuscripts were cited, but most did not indicate which % of hypohydration showed or not impairment on cognitive performance or impact on mental task load or mood. For example, Szinnai, Schachinger et al. 2005 showed cognitive-motor function is preserved during water deprivation in young humans up to a moderate dehydration level of 2.6% of body weight. Based on available literature, what basis did they have to suspect a difference in cognition, mental task load and mood at 2% body mass loss?

2) Population: In addition, it was not clearly explained the possible physiological differences in obese population that could affect these variables in comparison on non-obese individuals during or after water deprivation and/or heat stress. Why study this population regarding these aspects? One explanation brought is that military personnel, construction workers, and firefighters, are often exposed to occupational heat stress. Also, about one-third of Americans are reported to be overweight or obese. Please, clarify these point.

3) Hyperthermia: Possible explanation for thermoregulatory differences is that obese have a smaller BSA per unit of body mass than non-obese, which could affect their ability to effectively exchange heat with the environment. Since fat has a specific heat that is approximately half of that of fat-free mass, obese also can experience an insulative effect where a given heat load will increase their body temperature more than that of a lean individual (Buskirk et al. 1969). These possible differences between obese and non-obese are controlled in the present study as hydration levels and hyperthermia were the same between groups. Which physiological other physiological differences could affect cognitive responses, on mental task load and mood between groups? Please, include in the manuscript.

Specific Comments

4) Introduction. Line 73. However, other studies report no cognitive differences with mild hypohydration (~2%) when fluid is restricted for 36-h (Szinnai, Schachinger et al. 2005). Please, correct this sentence (one study or include other references).
In addition, how can this design lead to more hydration specific recommendations, when literature available suggests few or no effect will be detected at 2% body mass loss?

5) Methods. Line 101. Please, include additional information about sample size calculation.

6) Discussion. Line 325. What is the possible explanation for that a similar increase in body temperature resulted in an increased mental task in obese, but not non-obese?

7) The conclusion could be more of a take away message rather than an abrupt end more reflecting a result as it currently reads.

Reviewer 3 ·

Basic reporting

Please see comment in section 2 about references pertaining to head/brain heating and cognitive function.

Experimental design

1. N=21 (10 obese, 11 non); What kind of power did the authors have? Please provide this.
2. Does serum osmality/urine osmolality or Usg vary considerably during follicular phase?please comment.
3. “Learning effect” with ImPACT? (48 hr b/t trials) 5 parallel formats?
4. Replicated diet portion (calories) for subsequent trial… macro/micronutrient content could impact hydration state…? No mention was made by authors to address nutrient composition stated replication of diet (via caloric content)
5. “refrain from high moisture content foods” 24 hr prior in dehydration; was there significant difference prior to start of two trials? Why no baseline measures for Table 2, would they hold value to prove hydration levels against trial and not just pre/post a “control” so to speak ? Table 2 does not show statistical significance… were there no differences?
6. The water suit… head was not necessarily at same temp, could sweat while suit precludes sweat from body...this may be critical for cognitive study. What was the temperature of the brain? I know you did not measure this, but i needs to be addressed as your cognitive assessment in measuring brain function and we have no references to what this intervention does to the head/hippocampus/brain. This should be acknowledged as a limitation (also given the fact that the “global application” in the intro is around firefighters/military/athletes in their examples do not have their heads excluded from exposure).
7. Mood assessment after instrumentation/rectal probe, in suit? None taken on baseline day to determine if suit/positioning alone has impact on “mood” considering if instrumentation is a factor.

Validity of the findings

1. Statistical Analysis (Line 222): series of separate univariate ANOVA’s performed for all physiological (lists), hydration (lists). Reviewer notes this sentence is incomplete, believe word “variables” or similar may be missing.

2. What dose of heat increase occurred in brain? Please add any references if possible

Additional comments

1. Line 351 Capitalize Y in young

---

## Round 0.2 · accepted · Accept

Thanks for your effort in reviewing the manuscript.

# Reviewer 2 ·

Basic reporting

No comment.

Experimental design

No comment.

Validity of the findings

No comment.

Additional comments

The requested changes have been made. No additional comments.